# Investigation of Isotoxic Dose Escalation and Plan Quality with TDABC Analysis on a 0.35 T MR-Linac (MRL) System in Ablative 5-Fraction Stereotactic Magnetic Resonance-Guided Radiation Therapy (MRgRT) for Primary Pancreatic Cancer

**DOI:** 10.3390/jcm11092584

**Published:** 2022-05-05

**Authors:** Robert Hawranko, James J. Sohn, Keith Neiderer, Ed Bump, Timothy Harris, Emma C. Fields, Elisabeth Weiss, William Y. Song

**Affiliations:** 1Department of Radiation Oncology, Virginia Commonwealth University, Richmond, VA 23226, USA; hawrankore@vcu.edu (R.H.); keith.neiderer@vcuhealth.org (K.N.); ed.bump@vcuhealth.org (E.B.); timothy.harris@vcuhealth.org (T.H.); emma.fields@vcuhealth.org (E.C.F.); elisabeth.weiss@vcuhealth.org (E.W.); william.song@vcuhealth.org (W.Y.S.); 2Department of Radiation Oncology, Northwestern University Feinberg School of Medicine, Chicago, IL 60611, USA

**Keywords:** MR-Linac, MRgRT, dose escalation, pancreatic cancer

## Abstract

This study investigates plan quality generated by an MR-Linac (MRL) treatment planning system (TPS) for 5-fraction stereotactic body radiation therapy (SBRT) of primary pancreatic cancer (PCa). In addition, an isotoxic dose escalation was investigated with the MRL TPS based on stereotactic MR-guided adaptive radiation therapy (SMART) trial constraints. A clinical workflow was developed for adaptive and non-adaptive treatments with the MRL, on which a time-driven activity-based costing (TDABC) analysis was performed to quantify clinical efficacy. Fifteen PCa patients previously treated with a conventional Linac were retrospectively re-planned for this study. Three plans were generated for each patient using the original prescription dose (PD) and organ at risk (OAR) constraints (Plan 1), following SMART trial’s OAR constraints but with the original PD (Plan 2), starting with Plan 2, following an isotoxic dose escalation strategy where the dose was escalated until any one of the SMART trial’s OAR constraints reached its limit (Plan 3). Conformity index (CI) and the ratio of the 50% isodose volume to PTV (R50%) conformity metrics were calculated for all 45 MRL plans, in addition to standard dose-volume indices. Forty-five MRL plans were created which met their respective dosimetric criteria described above. For Plan 1, the MRL TPS successfully achieved equivalent or lower OAR doses while maintaining the prescribed PTV coverage for the 15 plans. A maximum dose to the small bowel was reduced on average by 4.97 Gy (range: 1.11–10.58 Gy). For Plan 2, the MRL TPS successfully met all SMART trial OAR constraints while maintaining equivalent PTV coverage. For Plan 3, the MRL TPS was able to escalate the prescription dose from the original 25–33 Gy by, on average, 36 Gy (range: 15–70 Gy), and dose to the PTV was successfully escalated to at least 50 Gy for all 15 plans. These achievements were made possible, in part, due to the omission of the ITV afforded by the MRL’s real-time target tracking technology and sharper dose penumbra due to its unique dual-focus MLC design. The 0.35T MRL TPS can generate plans that are equivalent to conventional Linac-based plans for SBRT of PCa. Through analyzing Plan 2 and 3 strategies, and due to the real-time target localization capabilities of the MRL system, increased OAR sparing and/or target dose escalation are possible.

## 1. Introduction

The American Cancer Society reported in 2000 that pancreatic cancer had a 5-year overall survival rate of less than 5%, even with aggressive management, and estimated it to cause 30,000 deaths per year in just the United States [1]. Findings from a more recent review of pancreatic cancer, published by the American Medical Association in 2021, which gathered from 43 randomized clinical trials 85 meta-analyses and 171 systematic reviews, showed the 5-year survival rate is approaching 10%, with approximately 60,000 novel cases per calendar year [2]. Although the 5-year overall survival has almost doubled over the last two decades, pancreatic cancer continues to hold its place as the lowest survival rate of any cancer. With current data indicating an increase in incidence of 0.5–1.0% per year, pancreatic cancer is projected to become the second-leading-cause of death from cancer by the year 2023 [2]. Currently the only curative treatment for pancreatic cancer is surgical resection, but only 15–20% of patients have resectable disease. While most unresectable disease is metastatic, 50–55% of total instance, there are a significant number of cases, 30–35% of total instance, that are considered to be locally advanced [2]. In the case of unresectable disease, radiation therapy offers palliative benefits, such as pain relief, and is often used in combination with other adjuvant therapies but is not used in an ablative capacity due to uncertainty in dosage to normal tissue as a result of motion. The first radiation therapy treatment for pancreatic cancer consisted of a conservative dose to the pancreas over a hyper-fractionated regimen; this technique is known as conventionally fractionated radiation therapy (CFRT). Stereotactic body radiation therapy (SBRT) is used to improve the outcome for pancreatic cancer patients undergoing an adjuvant radiation treatment. A review of 1147 patients treated with adjuvant external beam radiation therapy between the years of 2002 and 2014 revealed that the most common CFRT consisted of 50.4 Gy in 28 fractions delivered to the tumor and the most common SBRT consisted of 30 Gy in five fractions delivered to the tumor, and analysis of the data showed a statistical significance (*p* < 0.05) in 2-year overall survival for SBRT over CFRT, 26.9% and 13.7%, respectively [3]. The benefits of SBRT for unresectable pancreatic cancer is supported in the literature, but the technique is still restrained by the possibility of normal structure encroachment into the treatment site; thus, the total dose to the target volume is kept low enough to prevent early toxicity. The pancreas is particularly susceptible to motion due to its location in the abdomen. Peristalsis, breath motion, and day-to-day irregularities along with the extreme radiosensitivity of the gastrointestinal tract have trained physicists and oncologists to err on the side of caution when planning pancreatic SBRT. The introduction of onboard imaging, beginning with cone-beam computed tomography (CBCT) integrated with linear accelerator (Linac) was the first step toward solving this problem by allowing an on-table visualization of the daily anatomy, but CBCT images are not the best modality for soft tissue visualization, and they add ionizing radiation that patients receive. The most recent development in onboard imaging came in the form of magnetic resonance-guided radiation therapy (MRgRT). This technology has significant promise due to several factors: superior soft tissue contrast, real time anatomic information, active target tracking, and beam gating.

## 2. Materials and Methods

A collection of 15 PCa patients who were previously treated with a conventional C-Arm Linac were selected to be retrospectively re-planned for the 0.35 T MR-Linac (MRL) (ViewRay Inc., Oakwood Village, OH, USA) using the novel MRL’s commercial treatment planning system (TPS) (Version 5.4.0.97, ViewRay Inc., Oakwood Village, OH, USA). Three categories of plan were generated for each patient, starting from an original “conventional plan” that was used to treat the patient. The criteria for a patient to be selected for re-planning was that they had completed all external beam radiation treatment and that there was a 4DCT taken at the time of simulation. Plan 1 was created using the original prescription dose to the tumor volume and normal tissue dose constraints as used when the patient was planned and treated with a conventional TPS and Linac. A patient was randomly selected from the group and the data for Plan 1 is displayed in Figure 1A,D, which shows isodose lines at 100%, 50%, and 40% of Rx dose, which for this case is 30 Gy. Additionally, shown in this figure is the DVHs for critical organs at risk (OARs) such as small bowl, duodenum, stomach, GTV, and PTV. Plan 2 was created using the original plan’s prescription dose to the tumor volume but following a novel set of normal tissue dose constraints that were developed in a leading publication on stereotactic MR-guided adaptive radiation therapy, henceforth referred to as the stereotactic MR-guided adaptive radiation therapy (SMART) trial [4]. The organ at risk constraints outlined by the SMART trial can be found in Table 1.

The idea behind Plan 2 is to create a “robust” plan, which, in effect, means a plan that can remain within the normal tissue constraints listed in Table 1 without necessitating adaptation to daily anatomical changes. The same patient that was randomly selected for Plan 1 was also displayed for Plan 2 in Figure 1B,E, which shows isodose lines at 100%, 50%, and 40% of original Rx dose, which for this case is 30 Gy. Additionally, shown in this figure is the DVH for critical OARs such as small bowel, duodenum, stomach, GTV, and PTV. Plan 3 was created by taking Plan 2, then following an isotoxic dose escalation strategy wherein the dose was escalated until any one of the SMART trial’s OAR constraints reached its limit. The conventional Linac plans (25–33 Gy in 5 fractions) accounted for respiratory motion by creating an ITV that encompassed the GTV’s motion range, based on a 4DCT study before applying a PTV margin of 5 mm. Dose calculations were performed on a derived average intensity projection image. For the three plan types generated with the MRL TPS, the GTV was defined on the 0% phase, i.e., deepest inspiration phase, and was expanded uniformly in all directions with a PTV margin of 3 mm. The creation of ITV is unnecessary in this case due to the MRL’s real-time target tracking and gating technology, where in mild inspiration breath-hold (10% phase) treatments are allowed and implemented clinically. The same patent who was randomly selected for plans 1 and 2 above was also displayed for Plan 3 in Figure 1C,F, which shows isodose lines at 100%, 50%, and 40% of escalation dose, which for this case is 77 Gy. Additionally, shown in this figure are the DVHs for OARs, such as small bowel 0%, duodenum 0%, stomach 0%, GTV 0%, and PTV_high.

As mentioned above, an additional planning structure called PTV_high was created for Plan 3 and is defined as the PTV minus a 3 mm expansion of the OARs in close proximity to the PTV. A pass/fail criterion for Plan 3 evaluation is defined such that all of the following must be true: all SMART trial normal tissue dose constraints are met, PTV maintains 95% of original prescription dose, PTV_high maintains coverage of 95% of escalated dose, and PTV_high conformity index (Equations (1) and (2)) is less than or equal to 1.2. A typical strategy for creating Plan 3 was to begin by trying to escalate the dose to 50 Gy to 95% of the PTV_high volume. If it is not possible or reasonably feasible, the dose to the PTV_high is decreased at 5 Gy increments and re-evaluated at each iteration. If the 50 Gy escalation meets the criteria, the dose is escalated to the PTV_high. Depending on how easily the previous plan meets the criteria, this jump can be made by 5 or 10 Gy). This process is iterated until the plan meets all of Table 1 criteria, and the plan with the highest satisfactory escalation is accepted. Plan 3 was inspired by a growing body of literature that correlates high dose and low fraction treatment with increased local control in PCa. A comprehensive review of SBRT of the abdomen and pancreas was performed by the HyTEC group, the results of which lend strong support to increased local control after one year with hypofractionation [5]. Not only is there a positive correlation with hypofractionation, but data trends also suggest local control increases with dose escalation [5,6]. Local control is, in itself, a positive outcome and justifies further study into isotoxic dose escalation. However, recent studies have now started to build a case that increased local control leads to increased overall survival [6]. Indices for maximum and mean dose volume was calculated for several OARs on Plan 1 and was analyzed against the corresponding values from the original plan created for the patient on the conventional TPS.

An assessment of mean and max dose to typical OARs for the conventional TPS plans, which were created as volumetric modulated arc therapy (VMAT), and the MRL TPS plans, which were created as step-and-shoot IMRT, provides a point of reference for the shift in treatment technique. This is not meant as a direct comparison of VMAT and step-and-shoot IMRT. This analysis was not extended to plans 2 and 3 due to these plans having a novel set of normal tissue constraints and a prescribed dose to target, which would make the juxtaposition against originally planned parameters incoherent. It was necessary to use an alternate analysis on plans 2 and 3 against the conventional plan by comparing relative values such as conformity index (*CI*) and R50 conformity metrics. Equation (1), given as:(1)$$CI_{Plan\;2}=\frac{V_{RX\;isodose}}{V_{PTV}}\;$$
defines how *CI* is calculated for *Plan* 2 where *V_RX isodose_* is the volume of the prescription isodose line and *V_PTV_* is the volume of the *PTV*. Equation (2), given as:(2)$$CI_{Plan\;3}=\frac{V_{Escalation\;isodose}}{V_{PTV\_high}}\;$$
defines how *CI* is calculated for *Plan* 3, where *V_Escalation isodose_* is the volume of the dose that the plan was escalated to and *V_PTV_high_* is the volume of the *PTV_high*. Equation (3), given as:(3)$$R50_{plan\;2}=\frac{V_{0.5\;Rx\;Isodose}}{V_{PTV}}\;$$
defines how *R*50 is calculated for *Plan* 2, where *V_0.5 RX Isodose_* is the volume of the isodose line made by half of the prescription dose and *V_PTV_* is the volume of the *PTV*. Equation (4), given as:(4)$$R50_{plan\;3}=\frac{V_{0.5\;Escalation\;isodose}}{V_{PTV\_high}}\;$$
defines how *R*50 is calculated for *Plan* 3, where *V*_0.5 *Escalation isodose*_ is the volume of the isodose line made by half of the escalation dose and *V_PTV_high_* is the volume of the *PTV_high*.

In addition to a statistical analysis of plans created, a process map was created to chart simulation, nonadaptive, and adaptive treatment workflows, which can be found in Figure 2, Figure 3 and Figure 4, respectively. An analysis of the process map was carried out through a time-driven activity-based costing (TDABC) method in order to gauge quality and cost of treatment on the MRL, as well as inform process improvement. TDABC works by breaking down a technology into a step-by-step workflow, identifying which personnel interact with the workflow and for how long. A value is assigned to each individual’s time, and from this a cost analysis of the technology is derived. Our institution began treatment of patients on the MRL as of February 2022, and as such there was a limited amount of temporal data for our specific workflows. The data was tracked internally though log sheets broken down into the different steps shown in the process map. To quantify the cost involved with each personnel involved in the treatment workflow, a capacity cost rate (CCR) was calculated. In this study, CCR was determined by taking the average salary of the personnel and dividing by the time worked. The salary data was taken from ZipRecruiter.com (last accessed on 21 March 2022) and the time worked per year was determined using the total number of working days in 2021, i.e., 261 days, minus paid time off days, 43 according to VCU Health public data, then converting into the unit of minutes. These data are displayed in Table 2.

Note that these figures are based solely on average salary per year in Virginia according to a leading salary reporting service, and do not include bonuses or fringe benefits. Included in Table 2 is the cost rate description for a resident medical physicist, despite this position not appearing in any of the workflows. These data are included due to the medical physics resident performing IMRT QA in our department. For all intents and purposes this activity can be thought of as a single step between the simulation and planning workflow and either of the treatment delivery workflows, and on average takes 30 min to complete.

## 3. Results

Forty-five MRL plans were created which met their respective dosimetric criteria described above. For Plan 1, a statistical analysis of maximum and mean dose common OARs was carried out. A two tailed t-test (α = 0.05) failed to reject the null hypothesis that mean dose was no different for Plan 1 than for the conventional plan for spinal cord (*p* = 0.16), stomach (*p* = 0.49), duodenum (*p* = 0.48), and small bowel (*p* = 0.48). Further analysis showed that another two tailed t-test (α = 0.05) failed to reject the null hypothesis that maximum dose was no different for Plan 1 than for the conventional plan for spinal cord (*p* = 0.10), stomach (*p* = 0.48), duodenum (*p* = 0.47), and small bowel (*p* = 0.41). For Plan 2, the MRL TPS successfully created robust plans that met, without pushing, all SMART trial normal tissue constraints while maintaining equivalent PTV coverage. Plan 2 had an average CI of 1.05 (range: 0.98–1.24) and average R50 of 3.81 (range: 3.32–5.07) as depicted in Table 3. For Plan 3, the MRL TPS was successful in escalating the prescription dose from the original 25–33 Gy by, on average, 30 Gy (range: 15–70 Gy). Additionally, Plan 3 had an average CI of 1.13 (range: 0.99–1.40) and average R50 of 4.50 (range: 3.29–9.67), as depicted in Table 3. The dose to the PTV was successfully escalated to at least 50 Gy for all dose escalation plans, much beyond conventional fractionation of PCa SBRT, and it was even possible to escalate the dose to extreme values in patients with optimal normal tissue geometry. These achievements were made possible, in part, by the omission of an ITV. The creation of an ITV was deemed unnecessary due to the MRL’s real-time target tracking and gating technology, along with a mild inspiration breath-hold treatment protocol.

Through a TDABC analysis method, workflows have been generated that detail the personnel that interact with them and the extent of their involvement. Analysis of the simulation process (Figure 2) has shown that at least five types of staff interact directly with the process, the extent of which totals 80–160 min. The resulting personnel cost for simulation and planning is depicted in the data in Table 4 and was based on the average time spent per activity and totaled 129.17 USD. The nonadaptive treatment workflow (Figure 3) directly involves two types of staff and ranges from 40 to 110 min. The resulting personnel cost for the non-adaptive treatment is depicted in the data in Table 5 and was based on average time spent per activity, which totaled 71.12 USD. The adaptive workflow (Figure 4) is unsurprisingly more involved than that of the previous, and, as such, it involves four separate roles and takes between 68 and 156 min. The resulting personnel cost for adaptive treatment is depicted in the data in Table 6 and was based on the average time spent per activity which totaled 109.70 USD. Since each treatment is always predicated by simulation and planning, the total personnel cost for simulation, planning, and treatment is the sum of the two respective procedures. For non-adaptive planning, the end-to-end personnel cost is 200.29 USD. For adaptive planning, the end-to-end cost is 238.87 USD. An additional intermediate step for patients receiving treatment on the MRL is a CT simulation to provide electron density information required for dose calculation in planning. This was not accounted for in this study due to the authors’ choice to focus solely on costing from personnel on the MRL machine only.

## 4. Discussion and Conclusions

The 0.35 T MRL TPS can successfully generate plans that achieved equivalent OAR doses to those generated by conventional Linac-based treatments while maintaining prescribed PTV coverage. Through analyzing Plan 2 and 3 strategies, and due to the real-time target localization capabilities of the MRL system, increased OAR sparing and/or target dose escalation were found to be possible. The MRL system has proven, in this study, to be an effective platform for creating robust conventionally fractionated plans, as well as plans that safely escalate doses, which is anticipated to translate to possible clinical benefit for PCa patients. These achievements were made possible due to the MRL’s superior soft tissue contrast, real-time target tracking with beam gating, and sharper dose penumbra from its unique dual-focus MLC design. A TDABC analysis was carried out based on workflows for simulation and planning, non-adaptive treatments, and adaptive treatments, as well as a capacity cost rate determination for involved personnel. This analysis shows a 19.3% increase in personnel cost for adaptive treatment over non-adaptive treatment. Our next focus is to streamline the isotoxic dose escalation planning workflow for clinical implementation, as well as to explore various novel treatment strategies that play to the benefit of MRgRT.

## Figures and Tables

**Figure 1 jcm-11-02584-f001:**
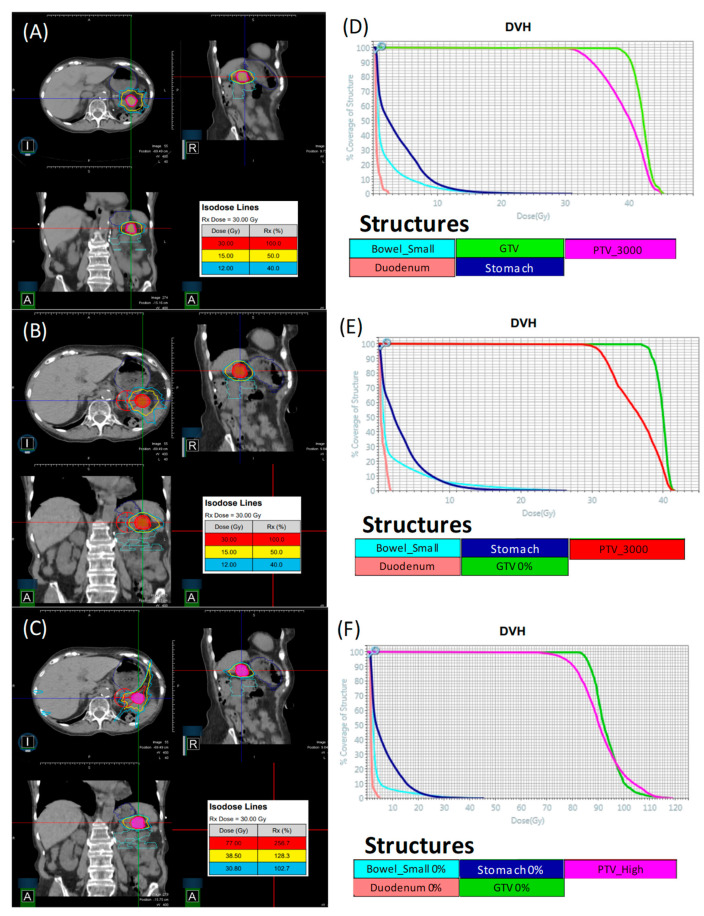
Depicts a sample patient picked from the group for Plan 1 (**A**,**D**), Plan 2 (**B**,**E**), and Plan 3 (**C**,**F**). Note that for this sample, Plan 3 escalation dose was 77 Gy to 95% of PTV_High.

**Figure 2 jcm-11-02584-f002:**
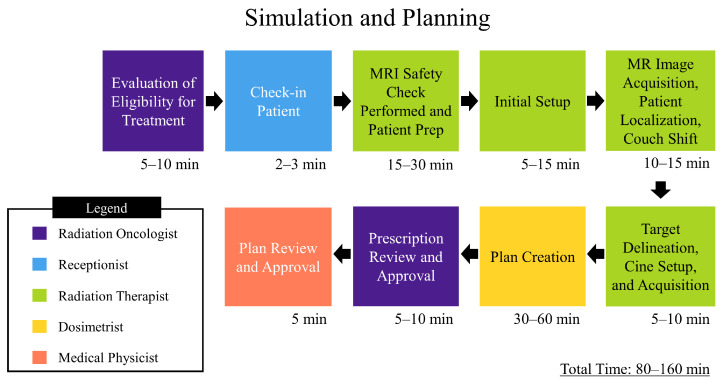
Depicts process map for MRgRT simulation. The times depicted are approximated from data gathered by VCU health staff.

**Figure 3 jcm-11-02584-f003:**
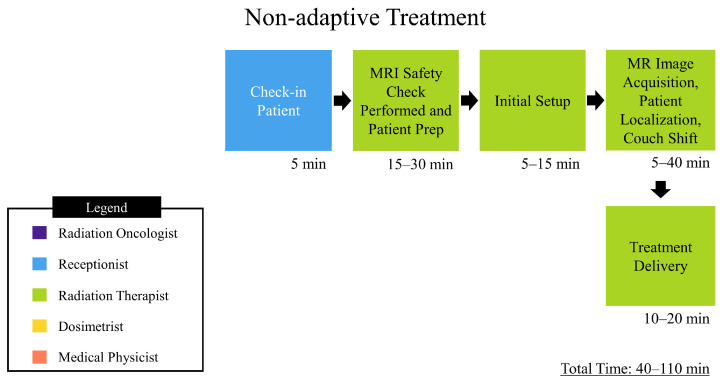
Depicts process map for MRgRT non-adaptive treatment. The times depicted are approximated from data gathered by VCU health staff.

**Figure 4 jcm-11-02584-f004:**
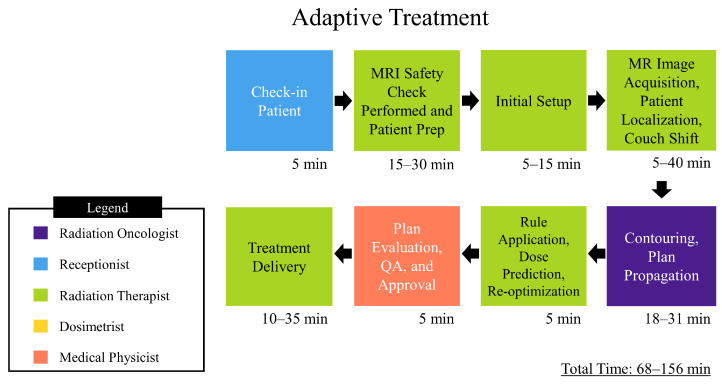
Depicts a process map for adaptive MRgRT. The times depicted are approximated from data gathered by VCU health staff.

**Table 1 jcm-11-02584-t001:** Depicts organ at risk (OAR) dose constraints as used in the stereotactic MR-guided adaptive radiation therapy (SMART) trial.

Organ at Risk	Dose Constraint
Stomach, duodenum, small bowel	V_35_ < 0.5 mL
	V_40_ < 0.03 mL
Large bowel	V_38_ < 0.5 mL
	V_43_ < 0.03 mL
Kidneys	D_mean_ < 10 Gy
Liver	D_mean_ < 15 Gy
Spinal cord	V_25_ < 0.03 mL

**Table 2 jcm-11-02584-t002:** Depicts the capacity cost rate information in USD for personnel involved in the MRL workflows. * Salary information for medical physics residents taken from publicly available VCU Health system house staff data for the 2021–2022 year.

Personnel	Virginia Avg. Salary [USD]	Cost Rate [USD/min]
Receptionist	30,545.00	0.292
Radiation Therapist	78,886.00	0.754
Radiation Oncologist	210,800.00	2.015
Dosimetrist	115,428.00	1.103
Medical Physicist	173,178.00	1.655
Resident Medical Physicist *	61,470.00	0.587

**Table 3 jcm-11-02584-t003:** Depicts average values and range of CI and R50 for plans 2 and 3.

Parameter	Plan 2	Plan 3
Avg. CI	1.05 (0.98–1.24)	1.13 (0.99–1.40)
Avg. R50	3.81 (3.32–5.07)	4.50 (3.29–9.67)

**Table 4 jcm-11-02584-t004:** Depicts the results of the TDABC analysis of the personnel cost associated with the simulation and planning workflow.

Personnel	Average Time Spent on Activity [min]	Cost [USD]
Receptionist	5	1.46
Radiation Therapist	52.5	39.58
Radiation Oncologist	15	30.22
Dosimetrist	45	49.64
Medical Physicist	5	8.27
	Total Cost:	129.17

**Table 5 jcm-11-02584-t005:** Depicts results of the TDABC analysis of personnel cost associated with the non-adaptive treatment workflow.

Personnel	Average Time Spent on Activity [min]	Cost [USD]
Receptionist	5	0.73
Radiation Therapist	70	52.77
Resident Medical Physicist	30	17.62
	Total Cost:	71.12

**Table 6 jcm-11-02584-t006:** Depicts results of the TDABC analysis of personnel cost associated with the adaptive treatment workflow.

Personnel	Average Time Spent on Activity [min]	Cost [USD]
Receptionist	5	1.46
Radiation Therapist	82.5	62.20
Radiation Oncologist	24.5	20.15
Medical Physicist	5	8.27
Resident Medical Physicist	30	17.62
	Total Cost:	109.70

## Data Availability

The data presented in this study are available on request from the corresponding author.

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
