# Peer review of "Investigation of Isotoxic Dose Escalation and Plan Quality with TDABC Analysis on a 0.35 T MR-Linac (MRL) System in Ablative 5-Fraction Stereotactic Magnetic Resonance-Guided Radiation Therapy (MRgRT) for Primary Pancreatic Cancer"

_jcm, 2022, doi:10.3390/jcm11092584_

Round 1

Reviewer 1 Report

The authors present a useful study of plan quality in the MRgRT setting with a particular focus on dose escalation – a topic of vital importance for pancreatic cancer patients – and include a time-cost analysis of MRI-guided workflows. The topics presented are valuable, but I have made a number of minor comments related to the readability of the manuscript.

Minor comments:

  1. Line 51: “With current data indicating represents…” Is this correct? The wording is not clear to me.
  2. Line 54: “respectable” -> “resectable”
  3. Line 60: Missing “cancer” after “pancreatic”?
  4. Line 100: “A collection of…” Starting at this line, sentences are just repeated from the beginning of the section.
  5. 2: “excalation” -> “escalation”
  6. Line 177: “and isodose line” seems misplaced.
  7. 3: Just a repeat of Eq. 2.
  8. Line 188: “Figures 1, 2, 3” -> “2, 3, 4”
  9. Line 218: “simulation” -> non-adaptive treatments
  10. Line 221: “non-adaptive” -> “adaptive”
  11. Line 246: “Figure 1” -> “2”
  12. Line 249: “Figure 2” -> “3”
  13. Line 252: “Figure 3” -> “4”
  14. Tables 3-5 should be referred to in the text.

Author Response

The authors present a useful study of plan quality in the MRgRT setting with a particular focus on dose
escalation – a topic of vital importance for pancreatic cancer patients – and include a time-cost analysis
of MRI-guided workflows. The topics presented are valuable, but I have made a number of minor
comments related to the readability of the manuscript.
Minor comments:
1. Line 51: “With current data indicating represents…” Is this correct? The wording is not clear to
me.
âž” Modified to not include “represents”.
2. Line 54: “respectable” -> “resectable”
âž” Modified.
3. Line 60: Missing “cancer” after “pancreatic”?
âž” Modified to read as “pancreatic cancer”.
4. Line 100: “A collection of…” Starting at this line, sentences are just repeated from the beginning
of the section.
âž” Thank you for pointing this out. Lines 105-122 were modified such that there is no repeated
information.
5. 2: “excalation” -> “escalation”
âž” Modified on line 185 and 186.
6. Line 177: “and isodose line” seems misplaced.
âž” Modified to exclude “and isodose line” on line 187.
7. 3: Just a repeat of Eq. 2.
âž” Modified on line 189.
8. Line 188: “Figures 1, 2, 3” -> “2, 3, 4”
âž” Modified to represent correct figures on line 198.
9. Line 218: “simulation” -> non-adaptive treatments
âž” Modified to read “non-adaptive treatment” on line 229.
10. Line 221: “non-adaptive” -> “adaptive”
âž” Modified to read “adaptive treatment” on line 232.
11. Line 246: “Figure 1” -> “2”
âž” Modified on line 260.
12. Line 249: “Figure 2” -> “3”
âž” Modified on line 264.
13. Line 252: “Figure 3” -> “4”
âž” Modified on line 267.
14. Tables 3-5 should be referred to in the text.
âž” Thank you for this suggestion. A new table has been added; therefore, these tables are now
listed as 4-6. Table 4 referred to in the text on line 262. Table 5 is referred to in the text on line
265. Table 6 is referred to in the text on line 269.

Reviewer 2 Report

General comments

This study is of potential interest to the radiation oncology community. However, there are many flaws in the manuscript.

Major comments

  1. The title of the manuscript and the statements in the introduction imply that this study was about comparing treatment plans of MRL treatment and a conventional treatment plan. The authors spend a good portion of the manuscript about the cost and efficiency of the treatment delivery. This topic does not match the title and introduction. Furthermore, the comparison of plans does not justify the work for a journal publication. I recommend the authors revise the title and introduction to present their experience with their new MRL machine. I am confident the new content could be suited for this journal.
  2. There are many editorial errors. Especially, the first two paragraphs in Materials and Methods contain duplicated sentences.
  3. The data for the plan comparison could also be presented in a table to better present the results.
  4. There are three plans. The authors mention the “conventional plan” in Results. Is it the fourth plan, and what is it?
  5. I am not sure if comparing VMAT and Step and Shoot technique is justifiable.

Minor comments

[Line 131] What is the “0%” phase? Is it the deepest inspiration phase?

Author Response

General comments:
This study is of potential interest to the radiation oncology community. However, there are many flaws
in the manuscript.
Major comments:
1. The title of the manuscript and the statements in the introduction imply that this study was
about comparing treatment plans of MRL treatment and a conventional treatment plan. The
authors spend a good portion of the manuscript about the cost and efficiency of the treatment
delivery. This topic does not match the title and introduction. Furthermore, the comparison of
plans does not justify the work for a journal publication. I recommend the authors revise the
title and introduction to present their experience with their new MRL machine. I am confident
the new content could be suited for this journal.
âž” Per your suggestion, the title has been modified to more accurately reflect the contents of the
manuscript.
2. There are many editorial errors. Especially, the first two paragraphs in Materials and Methods
contain duplicated sentences.
âž” Modified to fix editorial errors as described above.
3. The data for the plan comparison could also be presented in a table to better present the
results.
âž” Per your suggestion, Table 3 has been added in line 230 which reflects the conformity index and
R50 parameters tracked for plans 2 and 3.
4. There are three plans. The authors mention the “conventional plan” in Results. Is it the fourth
plan, and what is it?
âž” Thank you for pointing out this ambiguity. There are three plans that have been created by the
authors starting from an original “conventional plan” that was used to treat the patient, as this
was a retrospective re-planning study. This clarification has been included in lines 88-89.
5. I am not sure if comparing VMAT and Step and Shoot technique is justifiable.
âž” Thank you for bringing this to our attention. It was not the authors intentions to pursue a direct
comparison of VMAT and Step and Shoot techniques. The assessment of mean and max dose to
OARs between the two techniques is an aim at providing justification and a point of reference
for a shift in technique. This clarification has been added to the manuscript in lines 172-176.
Minor comments:
[Line 131] What is the “0%” phase? Is it the deepest inspiration phase?
âž” Thank you for posing this question. The 0% phase refers to the deepest inspiration phase, and
this clarification has been added to the manuscript in line 136.
Authors’ comments:
On behalf of my co-authors, we thank you very much for time and expertise in reviewing our article. We
have studied editor’s and reviewer’s comments carefully and have made revision which marked in
red/highlighted in the new manuscript. We have tried our best to revise our manuscript according to the
comments. Attached please find the revised version, which we would like to submit for your kind
consideration. We are grateful for your work to help us improve. 

Round 2

Reviewer 2 Report

Thank you very much for your responses. The manuscript improved and it is now satisfactory for me.